# “I’m Torn”: Qualitative Analysis of Dental Practitioner-Perceived Barriers, Facilitators, and Solutions to HPV Vaccine Promotion

**DOI:** 10.3390/healthcare12070780

**Published:** 2024-04-03

**Authors:** Meghan M. JaKa, Maren S. G. Henderson, Amanda D. Gillesby, Laura J. Zibley, Sarah M. Basile, Bryan S. Michalowicz, Donald Worley, Elyse O. Kharbanda, Steve E. Asche, Patricia L. Mabry, Brad D. Rindal

**Affiliations:** 1Center for Evaluation & Survey Research, HealthPartners Institute, Bloomington, MN 55425, USA; 2Center for Oral Health Integration, HealthPartners Institute, Bloomington, MN 55425, USAbryan.s.michalowicz@healthpartners.com (B.S.M.); donald.b.rindal@healthpartners.com (B.D.R.); 3HealthPartners Dental Group, Bloomington, MN 55425, USA; donald.c.worley@healthpartners.com; 4HealthPartners Institute, Bloomington, MN 55425, USA

**Keywords:** HPV vaccination, oropharyngeal cancer, semi-structured interviews, rapid qualitative analysis, implementation science, behavioral mechanisms

## Abstract

The human papillomavirus (HPV) vaccine can prevent HPV-related oropharyngeal cancers. Dental practitioners are uniquely positioned to promote HPV vaccines during routine dental care but experience barriers to doing so. Qualitative interviews were conducted with dental practitioners to understand barriers and inform intervention strategies to promote HPV vaccines. Dental practitioners were invited to participate in phone interviews about knowledge, self-efficacy, and the fear of negative consequences related to HPV vaccine promotion as well as feedback on potential interventions to address these barriers. Interviews were audio recorded, transcribed, and analyzed using rapid qualitative analysis with a sort-and-sift matrix approach. Interviews were completed with 11 practitioners from six dental clinics (avg. 31 min). Though most thought HPV vaccination was important, they lacked detailed knowledge about when and to whom the vaccine should be recommended. This led to a hypothesized need for discussions of sexual history, feelings of limited self-efficacy to make the recommendation, and fear of patient concerns. Still, practitioners were supportive of additional training opportunities and provided input into specific interventions. The nuance of how these barriers were described by practitioners, as well as the possible solutions they identified, will help shape future interventions supporting HPV vaccine promotion in dental care.

## 1. Introduction

Human papillomavirus (HPV) infection is increasingly common [1] and is a leading cause of oropharyngeal cancers [2]. The HPV vaccine is highly effective at preventing infection from the HPV strains that cause cancer if administered prior to exposure [3]. The 2- or 3-dose vaccine series is recommended from the age of 9 to 26 [4]. Unfortunately, in the United States, fewer than 60% of adolescents are adherent to HPV vaccination guidelines [5]. Given the relevance of oropharyngeal cancers to dentistry, preventive dental visits have been identified as an ideal setting for the promotion of HPV vaccines to prevent HPV-related oropharyngeal cancers [6]. In 2019, 87% of children and young adults visited a dental practitioner [7], providing an opportunity to reach many unvaccinated young people [8]. The National HPV Roundtable and dental professional organizations support the promotion of the HPV vaccine [9,10,11]. Oral disease prevention is within the scope of practice of dentists, dental therapists, and dental hygienists [6], but few currently promote vaccines in dental clinics [12]. Understanding the specific ways dental practitioners could engage with patients to promote the HPV vaccine is critical for designing effective training tools, education, and other interventions.

Foundational work has identified several barriers that prevent dental practitioners from promoting the HPV vaccine to patients. These barriers include practitioner knowledge of HPV, self-efficacy in promoting the vaccine to patients, and the fear of negative consequences (i.e., damaging the relationship with the patient or parent of a minor patient) [13,14,15]. Explorations of dental practitioner barriers to vaccine promotion have favored quantitative methods over qualitative and have focused heavily on knowledge with some attention to other barriers [15]. Consequently, initial interventions have focused largely on practitioner education [16,17]. Behavioral and implementation scientists have long encouraged the use of theoretical frameworks and qualitative methods to gain a deeper understanding of practitioner barriers beyond knowledge alone prior to designing intervention strategies [18,19,20]. Qualitative inquiry also allows an opportunity to gain dental practitioners’ perspectives on ways to address the barriers they face. Together, this information has the potential to lead to more effective and acceptable interventions to support dental practitioners in promoting the HPV vaccine to eligible patients. We conducted qualitative semi-structured interviews with dental practitioners in a large health system to understand barriers and inform the design of potential intervention strategies as part of a clinical trial planning process to develop and later test an intervention to support dental practitioners in promoting the HPV vaccine.

## 2. Materials and Methods

### 2.1. Setting and Population

Practitioner interviews were conducted in a large, integrated healthcare system based in Minnesota that offers insurance, medical and dental healthcare, education, and research [21]. Of the more than 90 clinics and hospitals [22] in the organization, dental care is provided at 21 locations. Within the system’s dental group clinics, there are over 130 dentists, dental hygienists, and dental therapists providing care to patients 9–26 years of age and, thus, eligible for the HPV vaccine. The organization uses an integrated electronic medical and dental health record system. Thus, dental practitioners using this system can access patient vaccine records and determine whether a patient is eligible for and/or overdue for one or more doses of the HPV vaccine series.

### 2.2. Interview Design

Semi-structured qualitative interviews were designed to elicit the barriers and facilitators faced by dental practitioners regarding HPV vaccine promotion, as well as to solicit feedback on possible interventions to enhance vaccine promotion. A semi-structured interview guide was designed to align with select barriers and proposed solutions [23] using the Capability, Opportunity, Motivation—Behavioral (COM-B) model and the Theoretical Domains Framework (TDF) as guides [24,25]. The three selected barriers of interest were knowledge of HPV and the HPV vaccine (capability), self-efficacy to make a strong HPV vaccine recommendation (motivation), and the fear of negative consequences of recommending the vaccine (motivation). While social and environmental factors (opportunity) are important in understanding vaccine promotion among dental practitioners (behavior), they were beyond the scope of this focused qualitative inquiry. The interview questions focused on proposed solutions that included enhanced practitioner training and tailored scripted messages to promote the vaccine during routine dental visits.

The interview guide was drafted using open-ended root questions that started with broader topics of disease prevention and funneled to more specific topics of HPV and HPV vaccination to prevent oropharyngeal cancers. The interview guide focused on key barriers to HPV vaccine promotion and suggested that facilitators promote HPV vaccination. Questions asked about participant knowledge, self-efficacy, and fear of negative consequences related to HPV and HPV vaccine promotion. Questions also asked about any previous experience respondents may have had promoting the HPV vaccine in dental practice settings. Each root question included several open-ended probes to ensure the correctness, clarity, and completeness of participant responses, avoiding bias and leading questions, and using neutral comments to facilitate the conversation and encourage depth in participant responses [26,27,28].

The drafted interview guide was then reviewed and revised by the full study team, which included vaccine experts, dental practitioners (dental hygienists and dentists), dental researchers, and implementation and behavioral science experts. The revised interview guide was pilot tested by trained qualitative interviewers at the Center for Evaluation and Survey Research (CESR) [29] at HealthPartners Institute with two dental practitioners on the study team, followed by a reflective conversation. Feedback was incorporated to improve the guide. The final guide was designed to take 30 min and included three brief introductory root questions to understand current care team practices and provider perspectives on the topic, two root questions related to the selected barriers to HPV vaccine promotion (knowledge, self-efficacy, and fear of negative consequences), one root question related to selected solutions, and a final closing question. See Appendix A for the final interview guide.

### 2.3. Interview Methods and Recruitment

Dental practitioners (i.e., dentists, dental therapists, and dental hygienists) providing routine care to patients ages 9–26 at the 11 clinics randomized to receive future intervention were identified through administrative electronic health record data. Of those eligible, practitioners were invited via both random selection (stratified by clinic and practitioner type) and dental leadership identification. Leaders were asked to identify practitioners who would be willing to share their perspectives on this topic. Once identified, practitioners were invited via a series of emails from dental leaders and study researchers. Initial invitation and reminder emails provided information about the interview opportunity, the confidential nature of the results, notification that a $50 gift card would be given to interview participants contingent upon interview completion, and a unique URL used to confirm eligibility to participate in the study interview, assess interest in participating, collect descriptive characteristics of the would-be respondent, and schedule the interview. Dental leaders also promoted the opportunity to participate in study interviews in weekly meetings and email communications.

Interviewers were from the same parent health system as the participating dental practitioners but were not otherwise known to the participants. In addition to facilitating pilot interviews, interviewers also reviewed HPV-related literature, attended HPV study-specific training, listened to interview recordings, and received ongoing supervision from a lead interviewer to ensure consistent data collection and reduce drift between interviewers. In addition, interviewers identified their own perspectives and biases and reflected on how to acknowledge and address issues of reflexivity in qualitative research [30]. At the time of the scheduled interview, trained and experienced qualitative interviewers called participants using phone/audio capabilities within Microsoft Teams software (version 1.0.1). At the start of each interview, respondents provided informed verbal consent, documented in the REDCap data collection file [31]. Throughout the interviews, interviewers took field notes to facilitate smooth transitions between topics and collect contextual data. All interviews were audio recorded and transcribed using Microsoft Teams software (version 1.0.1). Interview protocols were reviewed and approved by the organization’s institutional review board prior to data collection.

### 2.4. Qualitative Data Analysis

This qualitative research inquiry was conducted and reported following the COnsolidated criteria for REporting Qualitative research (COREQ) guidelines [32]. A target sample size of at least 10 completed interviews was planned based on the likelihood of saturation and given our research goals and sampling strategy [33]. Rapid qualitative methods were used to analyze interview data in alignment with the research questions of interest [34,35]. Immediately following each interview, interviewers trained in rapid qualitative methods created detailed memos [36] to summarize key points from the interview. Memos were recorded in REDCap under each interview guide section in alignment with the study’s a priori selected barriers to HPV vaccine promotion and/or the potential intervention targets. Then, following a sort-and-sift matrix analysis approach [37], a lead qualitative analyst reviewed recordings, transcripts, and memos creating a single matrix of interviews by constructs in NVivo v12 software (Denver, CO, USA). Rows and columns were then reviewed by a trained qualitative analyst (M.M.J.) to identify emergent themes within each construct, as well as key findings across and within interviews. Results were reviewed by a team of qualitative analysis and content experts on the study team to produce a final set of themes and again to consider issues of reflexivity and correctness [30]. As a final step, the lead analyst identified representative quotes to provide evidence of how the findings arose from the data [38] and summarized descriptive characteristics of participants using frequencies, percentages, means, standard deviations, minimums, and maximums.

## 3. Results

### 3.1. Descriptive Characteristics of Interview Respondents and Overview of Results

Twenty-four dental practitioners were invited to participate, of which eleven individuals (seven general dentists, one pediatric dentist, and three dental hygienists) from six different clinics agreed to participate and completed an interview. Participating practitioners had worked in this health system for an average of 12 years, with six having worked in the health system for less than 5 years and the rest having worked in the system for over 10 years. Five out of eleven identified as Black, Indigenous, or person of color, and six identified as White. Ten practitioners identified as female and one as male. Interviews lasted an average of 31 minutes (min: 27, max: 35 min) (see Table 1).

Almost all participating practitioners (10 of 11) reported never having recommended the HPV vaccine to patients. Two practitioners, unprompted, mentioned that they were aware of patient HPV vaccination status or eligibility in the dental health record, but only one practitioner regularly accessed this information and recommended the vaccine. When asked, five of eleven practitioners felt the vaccine should be recommended in dental practice. The remaining six had mixed feelings about whether it should be recommended or not with one practitioner feeling strongly the vaccine should not be recommended in dental practice. All practitioners described an array of barriers currently hindering their promotion of the vaccine. They also identified possible facilitators and intervention strategies that could support HPV vaccine promotion. Many practitioners reported mixed feelings about vaccine promotion and described inter-related barriers across categories, feeling torn between the importance of the topic and potential repercussions.


*“I’m torn because I think it’s a really good thing… we know there are links with oral cancer… So, I think it’s important… I think you’re going to find some hesitation, people may feel awkward talking about it…and the second thing would be the time.”*

*Dentist, #1018*


### 3.2. Dental Practitioner-Perceived Barriers Currently Preventing HPV Vaccine Promotion

Themes emerged across all three barriers explored: the lack of knowledge of HPV and the HPV vaccine (capability), the lack of self-efficacy to promote the vaccine (motivation), and the fear of negative consequences resulting from promoting the vaccine (motivation).

#### 3.2.1. Knowledge of HPV and the HPV Vaccine

With few exceptions, practitioners readily acknowledged a lack of knowledge and formal education regarding HPV. Many knew there was a link between HPV and oropharyngeal cancers but had little additional knowledge beyond that fact and had not received education on the topic as part of their dental training. Practitioners talked about learning that oropharyngeal cancer was primarily caused by smoking and alcohol habits.


*“We weren’t studying [HPV] when I was in school, so maybe they’re doing it now. Probably good to have it be part of the curriculum… Smoking causes cancer, we all know that.”*

*Dental Hygienist, #1020*


Practitioners also noted they had very little knowledge about the HPV vaccine including when and to whom it should be recommended. In the absence of this knowledge, practitioners began hypothesizing about the need to screen for risk of HPV exposure including asking patients about sexual history rather than making a universal recommendation. For example, in response to asking what information would be needed to make a vaccine recommendation, one dentist responded with the following:


*“… having pictures of what [HPV] would look like in different areas of the mouth would help just to kind of review… what to look for.”*

*Dentist, #1024*


#### 3.2.2. Self-Efficacy for Promoting the HPV Vaccine

The above knowledge gap and hypothesized need for a detailed risk assessment including a review of a patient’s sexual health history was often followed by a described lack of self-efficacy and a clear discomfort in making the vaccine recommendation.


*“I think some [colleagues] would feel uncomfortable talking about [the vaccine]… I just think, you know, it can be a sexual thing and there are people that just do not feel comfortable.”*

*Dental Hygienist, #1023*


The hypothesized need for a detailed conversation about risks or parent concerns also led many practitioners to describe concerns that they could not promote the HPV vaccine in a time-efficient manner. One hygienist described a lack of ability to promote the vaccine, even briefly, in the face of time-limited dental visits and competing priorities that require discussion during visits.


*“We just don’t have time. I know they say, ‘it’s just a sentence here or there’, but it all takes times and right now if you add up all the things we do in a visit, there is not enough time. We are always cutting something because we physically can’t get done what needs to be done in the time that we have. I think it throws people over the edge if you say, ‘here’s one more thing’.”*

*Dental Hygienist, #1023*


The barrier of limited time was also described as intertwined with experiencing any resistance from patients and in combination with other concerns about making the vaccine recommendation. One practitioner described the time needed to promote the vaccine with hesitant patients as not being worthwhile.


*“I’m not going to waste my time… I’ve argued with patients [about other topics before]… and I finally gave up… I’m done.”*

*Dental Hygienist, #1002*


#### 3.2.3. Fear of Negative Consequences from Promoting the HPV Vaccine

Related to the perceived patient hesitancy about the vaccine, practitioners described their fear of various negative consequences resulting from promoting the vaccine. One practitioner mentioned fear of patient complaints.


*“If I’m not required to do it, I just give up because I don’t want to have a [patient] complaint either… If you say the wrong thing to a parent or a kid or anybody, you could get a complaint.”*

*Dental Hygienist, #1020*


Others described the fear that patients may avoid seeking future care if they were to promote the vaccine in regular visits. Some also shared a perception that patients might not be comfortable talking about the topic with dentists.


*“I just have reservation… some patients are more open to disclose their [health] information to their medical doctor versus me, a dentist.”*

*Dentist, #1019*


### 3.3. Proposed Intervention Targets for Promoting the HPV Vaccine

When discussing facilitators or potential solutions to the described barriers, practitioners focused largely on logistic recommendations. Many described ideas to build the conversations into existing visit workflows, talking in detail about typical workflows, and the pros and cons of who and when to best promote the vaccine. One dentist described this team-based approach as a critical solution to address barriers like limited visit time.


*“Well, sometimes you really don’t have a choice but to make the time to communicate. You know, when the dentist needs to be moving ahead [so the dental hygienist or assistant can] finish up giving the recommendation… That has to be a team effort. Without that team effort, it does not happen.”*

*Dentist, #1009*


Regardless of role (i.e., dentist or dental hygienist), practitioners who voiced a lack of self-efficacy or a fear of negative consequences often suggested another staff member on their team would be better positioned to promote the HPV vaccine.

Another key logistic solution discussed by many practitioners was how to make recommendations to children under the age of 18 if the parent was not present in the exam room. Some, especially dental hygienists, described this as a current challenge with other dental recommendations and described complex solutions to contact parents in person, by phone, through electronic health records, or printed resources. Practitioners shared that some parents choose not to be present chairside at dental visits, instead opting to wait in the waiting room when a child is as young as 7 or 8 years old. By the time pediatric patients are 10–12 years of age, most parents are not present in the exam room during the dental exam.

Finally, practitioners described specific education, training, and supportive resources that may be effective in helping them promote the HPV vaccine. Many wanted additional information and education, but preferences on the mode and format of the education varied. Some described preferring short, self-guided information while others preferred in-person training where the whole clinic could hear the same message together to set joint expectations. Most practitioners also noted the importance of having a patient handout or resource, with some suggesting standard text that could be placed in a patient after-visit summary document. When asked, practitioners thought scripted messages could be helpful in responding to patient questions but acknowledged that in order for the message to feel authentic, the messages had to be flexible enough to accommodate the practitioner’s own communication style.

## 4. Discussion

Through qualitative interviews, dentists and dental hygienists in a large Midwestern US dental practice provided critically important, rich data on the barriers, facilitators, and intervention targets for promoting HPV vaccines in a dental setting. The current findings confirm and further extend previously published quantitative perspectives in the dental field [13,15]. The a priori barriers explored in these interviews were all present to varying degrees for these practitioners. The nuance of how these barriers were described by practitioners, as well as the possible facilitators and solutions they identified, will help shape an intervention targeting these barriers to HPV vaccine promotion.

Throughout these interviews, responding practitioners described knowledge, self-efficacy (especially in the face of limited visit time), and a fear of negative consequences as closely intertwined, suggesting that addressing one barrier alone may not lead to a successful intervention. As in other contexts [39,40], providing education on the course of HPV or the HPV vaccine may address practitioner knowledge gaps but may or may not build their confidence in promoting the vaccine among hesitant patients or alleviate their fears of negative consequences. Further, designing intervention strategies without a deeper understanding of these barriers is unlikely to be effective. By talking to practitioners in depth about the barriers they face, interventions can be designed that specifically and comprehensively support their needs. Using frameworks like the COM-B model helps to further specify and understand complex barriers and design strategies accordingly.

Another key point learned from these interviews was the importance of building HPV vaccine promotion strategies that fit within existing dental clinic workflows whether it be through the direct promotion of the vaccine or a referral to a healthcare provider. Working with practitioners and dental leaders is critical to developing a workflow that is flexible given the variability in how dental dyads or care teams work together. A successful intervention must allow care teams to play to one another’s strengths, ultimately allowing for a stronger HPV vaccine recommendation which has been shown to be predictive of vaccine uptake [41]. Similarly, workflow flexibility is needed when making HPV vaccine recommendations to minors. With many parents not attending dental visits for younger adolescents, and patients under 18 not being able to consent to vaccines in most states [42], promoting HPV vaccinations and writing orders for the vaccine becomes a key logistic challenge in dental care.


*Study Limitations*


Though this qualitative analysis of practitioner interviews provides critical information for intervention design, it is not without limitations. These data represent perspectives from a small sample of practitioners (N = 11) in one dental health system. Qualitative inquiries are not designed to create representative, generalizable knowledge, and rather are used to be exploratory or explanatory in nature [43]. While perspectives of both dental hygienists and dentists were heard through these interviews, another noted limitation is the lack of other clinic staff or direct patient voice in this work. As intervention strategies are designed, it will be critically important to gather patient feedback and perspectives as key stakeholders. The current work, together with the existing literature on best practices in the fields of dental care, vaccine research, and behavioral and implementation science can be triangulated to design an intervention that effectively addresses the barriers to vaccine promotion in routine dental visits. Interventions to support vaccine promotion and oropharyngeal cancer prevention in the dental office should carefully consider the barriers that currently prevent practitioners from promoting the HPV vaccine.

## 5. Conclusions

In conclusion, the work presented here elucidates the current barriers dental practitioners face when considering HPV vaccine promotion in routine dental visits: knowledge, self-efficacy, and fear. Semi-structured qualitative interviews allowed for a deeper nuanced understanding of each barrier, as well as the facilitators and potential intervention strategies needed to support vaccine promotion in this dental practice, such as individualized education opportunities and workflow flexibility. Future work should test the impact of such strategies on ameliorating the described barriers.

## Figures and Tables

**Table 1 healthcare-12-00780-t001:** Descriptive characteristics of the dental practitioners who completed semi-structured interviews, N = 11.

	N (%) or Mean (SD)
Practitioner Type	
Dentist	8 (73%)
Dental hygienists	3 (27%)
Years working in current health system	12 years (13 years)
Self-identify as Black, Indigenous, or another person of color	5 (45%)
Self-identify as female	10 (91%)
SD = Standard Deviation	

## Data Availability

Qualitative data will be made available upon request.

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
