# Peer review of "“I’m Torn”: Qualitative Analysis of Dental Practitioner-Perceived Barriers, Facilitators, and Solutions to HPV Vaccine Promotion"

_healthcare, 2024, doi:10.3390/healthcare12070780_

Round 1

Reviewer 1 Report

Comments and Suggestions for Authors

The authors performed a survey regarding HPV vaccination barriers among dentists. This is a well written manuscripts. But, it has some issues, mainly on the sample size. All concerns and suggestions are listed below:

1. while qualitative studies by interview sample size would usually only until thematic saturation, this study sample size is a bit small especially looking at the number of questions asked. It is stated the targeted sample size is 10, when they recruited 24 subjects initially. Why is that? seems conflicting. Kindly clarify.

2. The is no conclusion made at the end of the discussion. Only limitation and recommendation. Kindly add.

Author Response

The authors performed a survey regarding HPV vaccination barriers among dentists. This is a well written manuscripts. But, it has some issues, mainly on the sample size. All concerns and suggestions are listed below:

We appreciate the positive reactions and the constructive feedback which we feel has strengthened the manuscript. Please note we conducted phone interviews following the COnsolidated criteria for REporting Qualitative research (COREQ) guidelines.

1. while qualitative studies by interview sample size would usually only until thematic saturation, this study sample size is a bit small especially looking at the number of questions asked. It is stated the targeted sample size is 10, when they recruited 24 subjects initially. Why is that? seems conflicting. Kindly clarify.

Thank you for the question. The relatively modest sample size is a limitation of this work that we have noted in the discussion. We have further clarified our sampling approach, based on the research goals. We have added another reference that further describes the considerations we made in selecting the target sample size. Briefly, our goal was to interview 10 dental practitioners. In order to do this, we invited a convenience sample of 24 practitioners and 11 agreed to participate and completed interviews. 

2. The is no conclusion made at the end of the discussion. Only limitation and recommendation. Kindly add.

We have now added a conclusion paragraph summarizing our main findings and interpreting how they fit in the broader context.

Reviewer 2 Report

Comments and Suggestions for Authors

The Authors must see my remarks

Author Response

Article Type/Title: Please state the type of the article, eg. Research?

We have clarified this is an “original research manuscript” or Article, per the journal’s publication types. https://www.mdpi.com/journal/healthcare/instructions#submission

Reference for HealthPartners Institute’s Center for Evaluation and Survey Research

A reference has been added for HealthPartners Institute’s Center for Evaluation and Survey Research.

Tables are missing; more data is required; remove quotes from the article

A table containing descriptive characteristics of interview participants and interviews has been added in the results section. In qualitative inquiry, representative quotes are often used in the place of quantitative tables or figures to elucidate the depth of the data in a participant’s own voice. A key purpose is to demonstrate how the findings have arisen from the data. Following the COREQ guidelines, we have opted to leave these representative quotes in the manuscript. Per this suggestion, we have clarified this point in the methods section and added references to further support the qualitative research methods and practices.

Results will confuse the median readers, as statistical analysis is missing.

We followed COREQ guidelines for reporting the analysis section of this qualitative inquiry. We have also further specified the qualitative analysis section based on the reviewer’s suggestion. We have also added more detail about the descriptive statistics used.

Avoid repetitions… “The a priori barriers explored in these interviews lack of knowledge, lack of self-efficacy (especially in the face of limited visit time), and fear of negative consequences were all present to varying degrees for these practitioners.”

We edited this sentence to be less repetitious.

Personal opinion or Conclusions? “Knowledge, self-efficacy, and fear of negative consequences were closely intertwined for these responding practitioners, suggesting that addressing one alone would not lead to a successful intervention.”

This sentence was revised accordingly to be more closely drawn from the data.

Limitations?

We noted a number of key limitations in the manuscript including the modest sample size, the inclusion of a single health system, the lack of perspectives from dental clinic staff beyond dentists and hygienists, as well as the lack patient voice in this qualitative inquiry. This is now specifically identified via a limitation subheading.

Conclusion(s) Section?

We have now added a conclusions section.

Reviewer 3 Report

Comments and Suggestions for Authors

The manuscript is generally well-written, and the English is clear. It effectively communicates the purpose, methodology, and findings of the study. Overall, the discussion effectively communicates the key insights from the interviews and provides a strong foundation for future research and intervention development.

Author Response

The manuscript is generally well-written, and the English is clear. It effectively communicates the purpose, methodology, and findings of the study. Overall, the discussion effectively communicates the key insights from the interviews and provides a strong foundation for future research and intervention development.

We would like to thank this reviewer for the feedback.

Reviewer 4 Report

Comments and Suggestions for Authors

The authors present a compelling study that prompts reflection on the dentist's role in health promotion, extending beyond conventional dental practice. Their work is methodologically robust, delving into direct interactions with study subjects to gauge their perspectives on barriers, facilitators, and solutions related to promoting the human papillomavirus (HPV) vaccine. I have minimal observations on the paper, outlined below:

In the "Material and Methods" section, specifically lines 79 to 89, the authors employ the Theoretical Domains Framework (TDF) model and COM-B for designing the instrument to identify barriers. However, I noticed no identified barriers under the "Opportunity" category. While I understand the focus on motivation, could you clarify why no barriers were identified under "Opportunity," or if this was considered irrelevant in this model component?

Although the paper explores facilitators and potential solutions for HPV vaccination promotion, I couldn't find any mention of collaboration with specialists in the field within the instrument. While my observation might be anecdotal, in my practice, I've noticed an increase in referrals from dentists for counseling patients on HPV infection, investing between 30 minutes and an hour in this process. This made the paper particularly intriguing to me. I'm curious to know if the authors envision additional strategies for addressing this issue by involving specialists in the referral process.

Author Response

The authors present a compelling study that prompts reflection on the dentist's role in health promotion, extending beyond conventional dental practice. Their work is methodologically robust, delving into direct interactions with study subjects to gauge their perspectives on barriers, facilitators, and solutions related to promoting the human papillomavirus (HPV) vaccine. I have minimal observations on the paper, outlined below.

Thank you for the careful and thoughtful review of our manuscript.

In the "Material and Methods" section, specifically lines 79 to 89, the authors employ the Theoretical Domains Framework (TDF) model and COM-B for designing the instrument to identify barriers. However, I noticed no identified barriers under the "Opportunity" category. While I understand the focus on motivation, could you clarify why no barriers were identified under "Opportunity," or if this was considered irrelevant in this model component?

Thank you for noting this important point. We have further clarified that, although social and environmental opportunity are important determinants of behavior, they were beyond the scope of the current focused inquiry but are an important area for further research.

Although the paper explores facilitators and potential solutions for HPV vaccination promotion, I couldn't find any mention of collaboration with specialists in the field within the instrument. While my observation might be anecdotal, in my practice, I've noticed an increase in referrals from dentists for counseling patients on HPV infection, investing between 30 minutes and an hour in this process. This made the paper particularly intriguing to me. I'm curious to know if the authors envision additional strategies for addressing this issue by involving specialists in the referral process.

This is an important point and is especially relevant to treatment of HPV infection, whereas our focus is on prevention of infection through vaccination prior to exposure. However, even in primary prevention of infection through HPV vaccine promotion, referrals to other healthcare providers are an important tool and it is important for dental practitioners to be aware of vaccine-related resources relevant for their patient panels. We have added this to the discussion.

Round 2

Reviewer 1 Report

Comments and Suggestions for Authors

The authors address the concerns raised previously well. There are still inherent limitations of this study such as the small sample size but the authors did try to address this. No additional comments from this reviewer

Author Response

The authors address the concerns raised previously well. There are still inherent limitations of this study such as the small sample size but the authors did try to address this. No additional comments from this reviewer.

Thank you again for your careful and thoughtful review.

Reviewer 2 Report

Comments and Suggestions for Authors

In any case, and because I am a Reviewer for more than 520 International Medical Journals, I INSIST on the subject of Statistical Analysis

Morover, I can not accept parts of the text into the symbol: '' ... ''

The Authors must see my remarks

Author Response

Please state the type of the article, eg. Research?

This recommendation is in reference to a journal-specific heading that the authors are unable to change. If the editor finds it appropriate to change this from “Article” to “Research Article,” we would accept this designation.

[In reference to included qualitative data in the form of representative quotes] That style is unacceptable in scientific articles.....Remove....

We thank this reviewer for the recommendation and agree that qualitative research studies are less common than quantitative research studies in the healthcare field. Still, we think the Healthcare readership greatly value qualitative perspectives that center practitioner voice, particularly on this important special issue related to HPV and cancer prevention. Though qualitative research studies are less common in this field, we have provided some similar references below from this journal and our own work that include qualitative data in the form of direct, representative quotes:

·        https://www.mdpi.com/2227-9032/12/7/730

·        https://www.mdpi.com/2227-9032/12/7/717

·        https://www.mdpi.com/2227-9032/12/7/703

·        https://www.mdpi.com/2227-9032/12/6/685

·        https://journals.lww.com/jncqjournal/fulltext/2024/01000/the_role_of_care_coordination__a_qualitative_study.10.aspx

·        https://ajph.aphapublications.org/doi/full/10.2105/AJPH.2009.161364

Results will confuse the median Reader, as statistical analysis is missing....

As above, following the journal format for presenting qualitative research study findings, we have named this section Qualitative Data Analysis, parallel to a quantitative research study which would include a Statistical Analysis section.

In any case, and because I am a Reviewer for more than 520 International Medical Journals, I INSIST on the subject of Statistical Analysis

Morover, I can not accept parts of the text into the symbol: '' ... ''

The Authors must see my remarks

We thank the reviewer for their recommendation and hope the above references with examples from this journal will help to alleviate these concerns.

[In reference to the following sentence in the discussion, “The a priori barriers explored in these interviews related to knowledge, self-efficacy (especially in the face of limited visit time), and fear were all present to varying degrees for these practitioners.”] Avoid repetitions....

We have deleted the repetitious phrase in this sentence per the reviewer’s suggestion.

[In reference to the following sentence in the discussion, “For example, providing education on the course of HPV or the HPV vaccine may address practitioner knowledge gaps but may or may not build their confidence in promoting the vaccine among hesitant patients or alleviate their fears of negative consequences.] Personal opinions or Conclusions?

We revised this sentence to make clear that this is a point to consider in light of the results of this study and is supported by literature in other topical areas.